# Vernal Pool Amphibian Inventories in the Temperate Forests of Eastern North America: Can Environmental DNA Replace the Traditional Methods?

Bautisse Postaire *, Angélique Dupuch, Emilie Ladent and Yann Surget-Groba *

Département des Sciences Naturelles, Université du Québec en Outaouais, Ripon, QC J0V 1V0, Canada; angelique.dupuch@uqo.ca (A.D.); emilie_ladent@yahoo.com (E.L.)
* Correspondence: postaireb@gmail.com (B.P.); yann.surget-groba@uqo.ca (Y.S.-G.)

**Abstract:** Amphibian populations have been globally declining since at least 1990. In the temperate forests of eastern North America, vernal pools offer important breeding habitats to many amphibians but are usually not considered for forest management because of their small size and temporary status. The effective monitoring and management of amphibians rely on accurate knowledge of their spatiotemporal distributions, which is often expensive to collect due to the amount of fieldwork required. In this study, we tested whether eDNA metabarcoding could identify the same amphibian communities as the traditional inventory protocols. We collected eDNA samples in twelve vernal pools in the spring of 2019 and identified their communities via metabarcoding. At each pool, three traditional amphibian inventory methods were used: call surveys, trapping, and active search surveys. In total, 13 amphibian species were detected, with most of them being detected using both eDNA and the traditional methods. When comparing the results of eDNA with the traditional methods, we found that species ecology and behavior are key factors of its detectability via a specific method. With its higher taxonomical precision and repeatability, eDNA metabarcoding allows for the inventorying of amphibian species living and reproducing in vernal pools and their vicinity with accuracy. As eDNA metabarcoding is inexpensive compared to the traditional methods, we conclude that eDNA sampling should be considered for integration as a standard monitoring tool, after an initial assessment of amphibian diversity.

**Keywords:** amphibian; environmental DNA; metabarcoding; vernal pool; biodiversity inventory

## 1. Introduction

The 15th Goal of the Sustainable Development Goals of the United Nations stipulates that the protection and restoration of forest ecosystems is a critical challenge for all nations [1]. Healthy forest ecosystems and the biodiversity they harbor are crucial, as they provide multiple irreplaceable ecosystem services [2,3]. However, as all ecosystems, forests are facing major biodiversity losses, mainly due to human activities [4,5]. In many countries, public authorities and the private sector are therefore requested or incited to monitor forest biodiversity and report their management efforts, e.g., in the framework of the Convention on Biological Diversity or the Sustainable Forestry Initiative. However, forest management practices impact taxonomic groups in various ways [6], and affordable tools able to detect early changes in biodiversity are often lacking, impacting the reports' accuracy and management decisions [7].

Among other forest-associated taxa, amphibian populations have been following a downward trend globally for at least 30 years, resulting from a conjunction of several environmental threats [8–10]. Effective assessments, monitoring, and management of these organisms rely on accurate knowledge of their spatiotemporal distributions and abundance, which is often expensive to assess due to the amount of fieldwork required to obtain reliable and representative data. Due to their various habitats and behaviors, amphibian inventories

historically rely on the combination of different techniques, including traps and call or active search surveys [11]; these require extensive training and are difficult to scale over large and/or remote areas. There is a need for cost-efficient and fast alternatives to accelerate the understanding of amphibian distribution, as well as their populations' temporal dynamics.

Environmental DNA detection has emerged over the last decade as a new approach to detect organisms in water bodies [12–14]. This technique relies on the capture, amplification, and detection of DNA fragments from a water sample, originating from the cells left in the environment by species of interest. Non-invasive, field cost-efficient, and requiring a limited expertise of the studied organisms, eDNA detection represents a promising addition to amphibian monitoring and inventories. Since the first application of eDNA on animals, which was to detect invasive bullfrogs in France [15], this approach has been successfully used multiple times to detect various amphibians and many other organisms [16–19]. Most amphibians rely on freshwater streams and waterbodies for reproduction, producing significant amounts of eDNA in the process, easing their detection. However, some species rely on temporary and small waterbodies or use them for short periods [20], in which eDNA accumulation may not be enough to ensure reliable detection, even though the species is present in the area. This is pronounced for species inhabiting higher latitudes and are exposed to a continental or Mediterranean semi-desertic climate, where temporary waterbodies form during spring, which are essential for reproduction [20,21]. These habitats, called vernal pools [22,23], are essential for several amphibians as they provide a predator-free environment due to the absence of fish [24,25]. While a common feature of North-Eastern American temperate forests [23], these habitats have been extensively impacted by forest management activities since the beginning of the 20th century [22,23,26,27], and active monitoring is necessary to assess the health of animal populations which rely on them.

According to the Atlas des amphibiens et des reptiles du Québec [28], 21 Urodela and Anura species are present in Québec (Canada). Most of them rely on local forests and natural waterbodies such as vernal pools to maintain their populations [23,29]. Developing a monitoring tool utilizing eDNA could aid long-term conservation by precisely assessing distribution ranges and their potential changes. However, the short-lasting nature of vernal pools throughout the year could pose a problem for their eDNA detection. Indeed, it depends on several factors including, among others, organisms' activity, environmental variables, the density of the population, etc. [30–32]. Notably, recent studies managed to track the temporal variation in amphibians in vernal pools utilizing eDNA, but highlighted that the detections were dependent on the species ecologies and environmental conditions [16,33–35]. If successful, this flexible approach would allow the monitoring of several taxonomic communities (amphibians, arthropods, annelids, etc.) using a single lab procedure, drastically reducing lab costs in comparison with a traditional approach.

In this study, we thus aimed to test whether eDNA metabarcoding could identify the same amphibian communities as a combination of the traditional inventory protocols in the vernal pools of southwestern Quebec (Canada), expanding further the body of literature on the subject. Based on the previous studies [16,35], we predicted that the diversity detected via eDNA amplification would be at least on par with the traditional inventory methods, as some amphibians do not sing and are difficult to find (e.g., Urodela), allowing an easier and faster assessment of their presence compared to the traditional methods.

## 2. Materials and Methods

### 2.1. Sampling Site

The Kenauk Reserve (45.75°, −74.81°) in Quebec is a privately owned and manages temperate forest area (~265 km$^2$) situated between 50 and 400 m above sea level. It comprises around 60 lakes and several hundred vernal pools [36], with tree cover consisting of typical local deciduous species (*Acer saccharum*, *Betula alleghaniensis*, *Fagus grandifolia*, and *Quercus rubra*) and conifers (*Tsuga canadensis*). Parts of this domain have been exposed to various degrees of wood extraction over the last century, from being untouched to clear

cuts, providing a wide range of habitats and disturbances; a valuable set up to test wildlife sampling and monitoring methods. For this study, we randomly selected 12 vernal pools across Kenauk Reserve [37]. We conducted amphibian diversity inventories and sampled eDNA between May and June 2019 (Figure 1, Table S1). Based on previous inventories, 13 species of amphibians have been observed in the area [38].

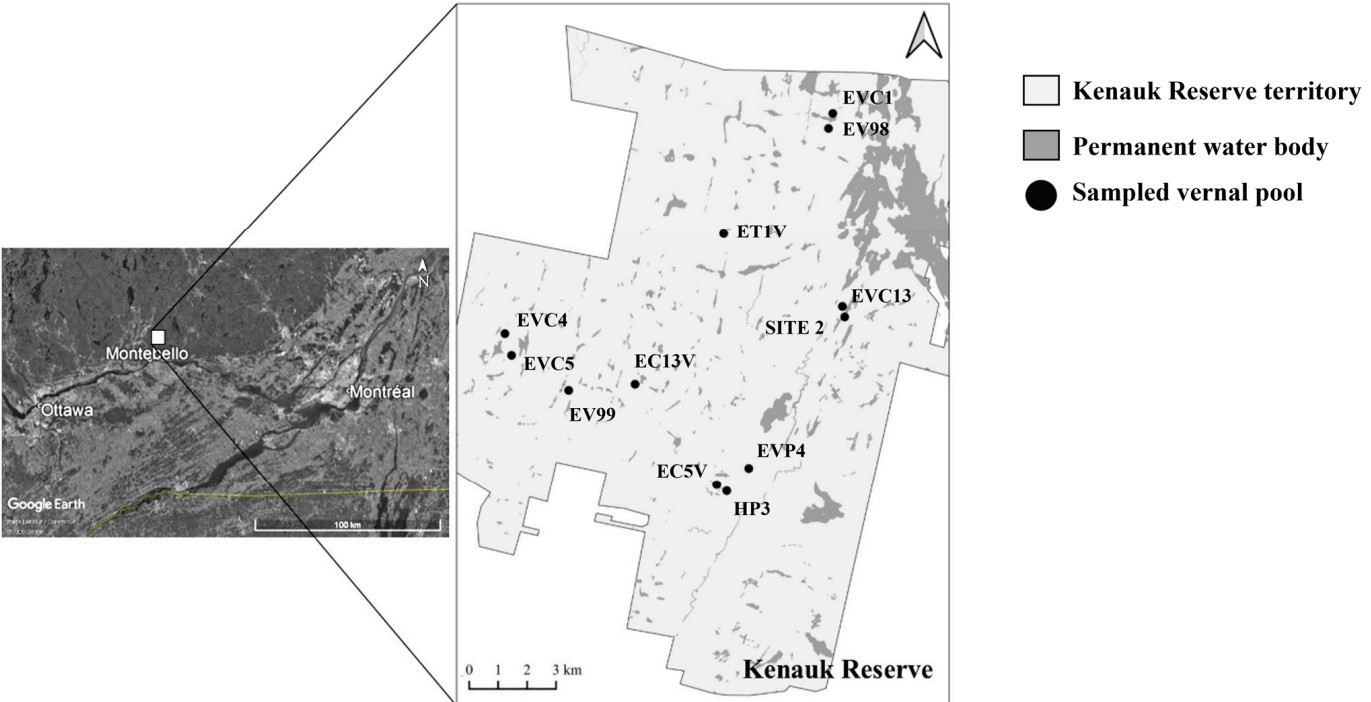

**Figure 1.** Positions of sampled vernal pools in Kenauk Reserve (QC, Canada). Vernal pools' names are also indicated. Based on [29].

### 2.2. Traditional Inventory of Amphibian Diversity

Three complementary inventory methods were used to assess amphibian diversity within and in the vicinity of each sampled vernal pool.

First, a call survey was conducted by planting an acoustic recorder (Swift–Terrestrial Passive Acoustic Recording Unit, The Cornell Lab) on each sampling site (on the closest available tree to the vernal pool). Every three days, calls were recorded 5 min/h between 9 p.m. and 11 p.m., throughout the sampling period (total: 24 h sampled between 12 May and 2 June; Table S1). Recordings were analyzed using Audacity® Cross-Platform Sound Editor and Anura calls were identified down to the species. When call recordings could not be assigned undoubtedly to a species, they were sent to an external expert (Sébastien Rouleau, coordinator of the Atlas des Amphibiens et Reptiles du Québec) for confirmation.

Second, five steel mesh Gee traps were set in each sampled vernal pool (pool noodles were placed inside the trap to ensure access to breathing space and to prevent drowning) for four to six consecutive days (Table S1) and checked every day (total trapping days: 68). Adult amphibians were identified down to the species and released after each trap check.

Finally, three active search surveys were conducted in a 20 m radius around each vernal pool: on the day when the steel mesh Gee traps were installed, after each trap check, and on the day of trap removal. Each survey was independently conducted by two people and consisted of 10 min random walks during daylight (total time of active search surveys: 16.3 h), checking under rocks and wood logs if present. Encountered amphibians were identified down to the species.

## 2.3. eDNA Collection and Processing

Water samples were collected concomitantly to inventories. Six to nine water samples were collected in total per vernal pool (Table S1), to cover a range of at least 28 days (4 weeks) for each pool, with a maximum of 17 days between two sampling events (average: 9.8 days). On each sampling occasion, one or two water samples were collected. Water samples were collected in mid water, taking care not to stir up sediment, thus limiting sediment capture and the ancient eDNA it may contain [39].

A total of 95 water samples of 1L each were filtered immediately on site with a peristaltic Masterflex® L/S® Portable Sampling Pump (Avantor, Radnor, PA, USA). A cellulose nitrate membrane and their disposable holders (0.47 μm pore size; Nalgene® Analytical Test Filter Funnels, ThermoFisher Scientific, Waltham, MA, USA) were stored in a cooler with ice packs for transport to the laboratory. Using clean (1% bleach and 30 min UV exposition) tweezers, the filters were transferred in sterile vials containing 5ml of pure ethanol and stored at −70 °C until extraction. The cooler and tweezers were decontaminated using a 1% sodium hypochlorite solution before and after use. DNA extractions from whole filters were performed using the DNeasy Blood and Tissue Kit (Qiagen, Germantown, MD, USA), following the manufacturers' protocols except for the modifications described in [40].

## 2.4. Amplicon Preparation and Sequencing

Using the preliminary data, we confirmed that primers MHemF and dgHCO2198, which were used previously for the metabarcoding of arthropod communities [41], successfully amplify vertebrates. However, given the high abundance of several invertebrate taxa in the vernal pools, most of the generated reads belonged to annelids or arthropods. We thus designed new primers preferentially targeting vertebrates to try to improve their representation in the final dataset. We thus used two different amplicons for this study: a 455 bp fragment amplified using primer MHemF/dgHCO2198, and a 176 bp fragment amplified using the new primers, Rept176F (GGRGCHATYAAYTTYATYAC) and Rept176R (GWKGTRTTNAKRTTTCGRTC). PCR were conducted in 20μL reactions containing 0.2 μM of each primer and 10 μL of Qiagen Multiplex PCR Kit. Cycling parameters followed the Qiagen Multiplex PCR Kit recommendations, using a 50 °C annealing temperature. The two amplicons were then pooled for each sample and sent to the Centre d'expertise et de services Génome Québec for sequencing in an Illumina MiSeq sequencer using a MiSeq Reagent Kit v2 (500 cycles, Illumina, San Diego, CA, USA).

## 2.5. Contamination Control

All field and laboratory steps followed thorough contamination control procedures. Disposable gloves were used in the field, changing between each sample collection. Water samples were taken 10 cm below the water surface to limit collecting sediment, and on the edge of each pond to prevent sediment resuspension due to the water movements. Filter funnel-holders were cleaned and decontaminated after use with a 1% sodium hypochlorite solution. Extractions, library preparation, and sequencing were physically and temporally separated using dedicated lab equipment. All eDNA extractions and PCRs were conducted in the molecular ecology lab at ISFORT, inside different dedicated hoods; all laboratory equipment, including hoods, was cleaned with a 1% sodium hypochlorite solution and subsequently exposed to a 15 min UV sterilization before and after each use. Disposable gloves were used, changing at minima between each sample. Filters were cut using stainless scissors, directly inside the DNA digestion tubes, with sterilizing with flame between samples. All work surfaces were cleaned with a 1% sodium hypochlorite solution prior to and after each use, as was the equipment used for extraction. To identify potential contaminations, negative controls were added to the protocol by filtering 1L of ddH$_2$O in the field, and by the use of PCR blanks.

### 2.6. Metabarcoding: Bioinformatics and Taxonomic Assignment

The Illumina sequencing data was demultiplexed using the Illumina MiSeq software [42]. Individual FASTQ files were then filtered and cleaned following a series of quality control steps using Usearch v11.0.667 [43]. The parameters of the pipeline for each fragment can be found in the Text S1. A local database (a compilation of 12,642 CO1 sequences of Chordata, Arthropoda, Mollusca, and Gastropoda present in the sampling area) was used to assign zOTUs with a taxonomical identification using the -sintax command in Usearch, keeping only the ranks with a confidence assignment >0.8. zOTUs per fragment were mapped to the database sequences with the -otutab command. zOTUs assigned to the same Amphibia species were merged (only two species were represented by more than one zOTU); zOTU sequences were also manually submitted to BOLD [44] to confirm identification and to identify unassigned sequences using -sintax. We obtained a list of species per vernal pool detected via eDNA sampling, including their number of reads. The results of other inventory methods (Table S2) were merged to create a list of species per vernal pool detected via traditional sampling.

### 2.7. Statistical Analysis

All statistical analyses were performed in R v 4.1.2, mainly with the vegan v 2.6-2 and ggplot2 v 3.4 packages [45–47]. The comparison of amphibian communities obtained from the two inventory methods (eDNA versus the combined traditional approaches) was based on the two presence/absence matrices of the species. First, a Venn diagram was drawn to visually represent the differences in the number of species detected between the two methods. Then, to determine whether the sampling effort was sufficient with either method, the species accumulation curves of the number of species detected as a function of the sampled vernal pools were plotted with the specaccum() function, performing 1000 random permutations. For both methods, the Chao2 estimator [48], which estimates the number of species theoretically detectable based on the proportion of species observed once or twice in the dataset, was calculated using the package fossil v 0.4.0 [49]. A Mantel test was performed to assess the correlation of matrices measuring Sorensen's dissimilarity index [50] between the sampled vernal pools using either the inventory method; this test measures if communities' dissimilarities among vernal pools follow the same trends when using the traditional methods or eDNA sampling. Subsequently, a permutational multivariate analysis of variance (PERMANOVA) was also performed, with Sorensen's dissimilarity index and 1000 permutations using the adonis() function, to test the similarity of the retrieved communities using either inventory method. Finally, the correlation between the number of vernal pools where a species was detected using eDNA sampling or the traditional approaches was tested to identify if the methods had the same detection power for each species.

## 3. Results

### 3.1. Metabarcoding Data Analyses

The Illumina sequencing runs generated a total of approximately 5.74 million and 0.75 million filtered paired-end reads for the 176 bp and 455 bp fragments, respectively. Negative control PCRs generated few or no reads, suggesting that insignificant contamination was observed during the library preparation and Illumina sequencing. After initial filtering, we identified 1018 zOTUs with the 176 bp fragment and 1135 zOTUs with the 455 bp, among which 13 and 6 zOTUs were identified as Amphibia, respectively. Both barcodes yielded the same taxonomic resolution, as all Amphibia zOTUs were identified down to the species level. The 455 bp fragment failed to detect *Ambystoma laterale* (Hallowell 1856), *Anaxyrus americanus* (Holbrook, 1836), *Plethodon cinereus* (Green, 1818), *Pseudacris crucifer* (Wied-Neuwied, 1838), *Lithobathes clamitans* (Latreille, 1801), *and Lithobathes palustris* (Le Conte, 1825). Due to the varying success of both fragments, we decided to group all detections using eDNA, regardless of the CO1 fragment, and compare them to the traditional sampling methods.

In total, 11 amphibian species were detected across the sampling area using eDNA metabarcoding (Table 1). The amphibian species accumulation curve reached an asymptote after approximately 50 samples, when considering the collected filters independently over the sampling area (Figure S1), indicating that the sampling effort was sufficient to detect most species that this method could uncover over the sampled area.

**Table 1.** Number of sites where amphibian species were detected using eDNA, acoustic recorder (Acco), Gee traps (Trap), and active search surveys (Enco) and sampling methods over the investigated period. Amphibian species are sorted in alphabetical order per taxonomic family. Asterisks indicate Morphos species previously detected in Kenauk Reserve [30] among species recorded in Quebec [20].

| Order | Family | Species | eDNA | Acco | Trap | Enco |
|-------|--------|---------|------|------|------|------|
| Anura | Bufonidae | *Anaxyrus americanus* * | 5 | 1 | 0 | 0 |
| | Hylidae | *Hyla versicolor* * | 4 | 0 | 0 | 0 |
| | | *Pseudacris crucifer* * | 4 | 10 | 1 | 0 |
| | | *Ps. maculata* | 0 | 0 | 0 | 0 |
| | | *Ps. triseriata* | 0 | 0 | 0 | 0 |
| | Ranidae | *Lithobates catesbeianus* * | 4 | 0 | 0 | 1 |
| | | *L. clamitans* * | 12 | 3 | 8 | 7 |
| | | *L. palustris* * | 1 | 0 | 0 | 0 |
| | | *L. pipiens* | 0 | 0 | 0 | 1 |
| | | *L. septentrionalis* * | 0 | 0 | 0 | 0 |
| | | *L. sylvaticus* | 1 | 4 | 0 | 1 |
| Urodela | Ambystomatidae | *Ambystoma laterale* | 3 | 0 | 0 | 0 |
| | | *Am. maculatum* * | 9 | 0 | 0 | 2 |
| | Plethodontidae | *Desmognathus fuscus* | 0 | 0 | 0 | 0 |
| | | *D. ochrophaeus* | 0 | 0 | 0 | 0 |
| | | *Eurycea bislineata* * | 0 | 0 | 0 | 0 |
| | | *Gyrinophilus porphyriticus* | 0 | 0 | 0 | 0 |
| | | *Hemidactylium scutatum* * | 0 | 0 | 0 | 1 |
| | | *Plethodon cinereus* * | 2 | 0 | 0 | 12 |
| | Proteidae | *Necturus maculosus* | 0 | 0 | 0 | 0 |
| | Salamandridae | *Notophthalmus viridescens* * | 6 | 0 | 0 | 4 |

*3.2. Comparison between Traditional Inventory Methods and eDNA Sampling*

In total, 13 amphibian species were detected when aggregating all sampling methods (10 for the traditional methods and 11 for eDNA metabarcoding; Table 1). Each traditional method yielded different results in terms of species composition (from two species with Gee traps to eight for active search surveys; Table 1). Among the detected species, *L. clamitans* was the most common (Table 1), as detected in all vernal pools by at least one traditional inventory method as well as eDNA sampling. The Venn diagram revealed that both methods detected most species (eight species), with only two and three species detected only via traditional or eDNA sampling, respectively (Figure 2). Interestingly, three species were reported for the first time in Kenauk Nature (i.e., *Lithobates pipiens* (Schreber, 1782), *Lithobates sylvaticus* (LeConte, 1825), and *Ambystoma laterale*), while two previously reported (*Lithobathes septentrionalis* (Baird, 1854) and *Eurycea bislineata* (Green, 1818)) were not detected.

**Figure 2.** Venn diagram of amphibian species detected using traditional (TS) and eDNA sampling. Asterisks indicated species previously reported in Kenauk Nature [38].

Species rarefaction curves showed that when considering the vernal pools as sampling units, the number of sites sampled (n = 12) was insufficient to capture all species diversity for either eDNA or the traditional inventory surveys (Figure 3). Chao2 indices indicated that traditional sampling had the potential to detect more species than eDNA. They also showed that traditional sampling was missing almost half of the theoretical diversity, with a high proportion of species detected only once. On the contrary, the total number of amphibian species detected using eDNA was close to its Chao2 estimate, indicating that most species were detected several times. The non-significance of the Mantel test ($p$ value = 0.0538) indicated that the site dissimilarities matrices of the eDNA and traditional inventories were uncorrelated, underlining that communities' composition patterns were different across the sampled area based on the inventory method. Furthermore, the composition of amphibian communities clearly differed between inventory methods (PERMANOVA, $p$ value < 0.01), with eDNA communities being significantly more similar (Wilcoxon test $p$ value < 0.001). This discrepancy was especially clear with the uncorrelated number of detections per sampling method for each species (Spearman's rank correlation $\rho$ = −0.079, $p$ value = 0.56). In other words, some species had different detection rates depending on the method employed (e.g., *Pl. cinereus*, *Ps. crucifer*, *Ambystoma maculatum* (Shaw, 1802); Figure 4).

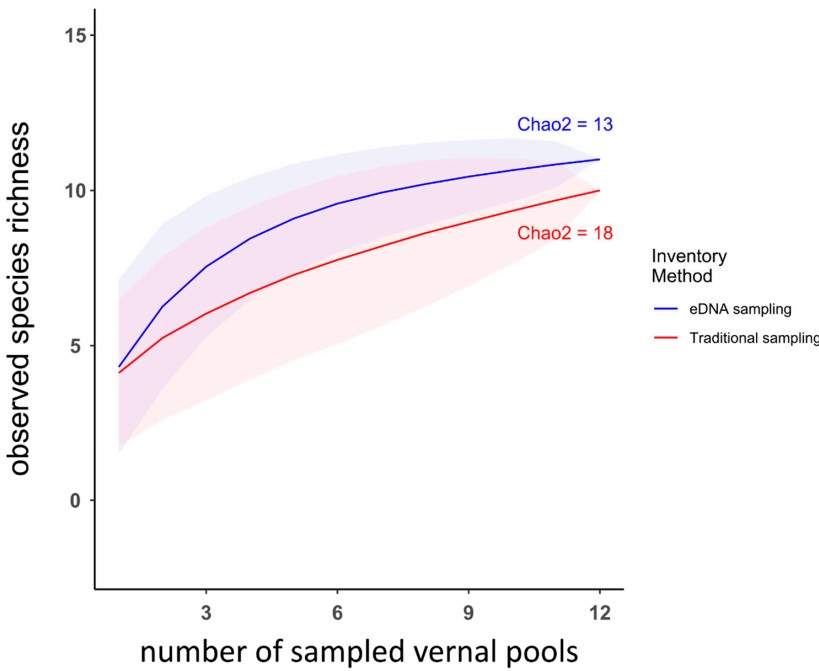

**Figure 3.** Amphibian species' accumulation curves as a function of the number of sampled vernal pools, detected using eDNA metabarcoding (blue) or traditional sampling methods (red). Transparent areas correspond to the 95% confidence intervals, estimated from 1000 random permutations. The Chao2 estimator of maximum expected species richness via inventory approach are also indicated.

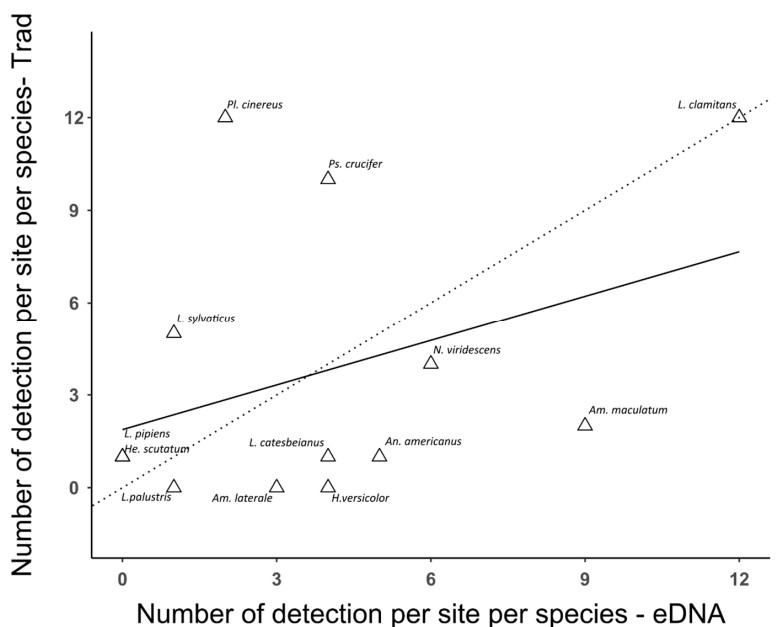

**Figure 4.** Number of detections per site per amphibian species using either traditional sampling (Trad; *y*-axis) or eDNA sampling (eDNA; *x*-axis). Only amphibian species detected at least once by either method are shown. The dotted line represents the perfect relationship (1:1) between the two inventory methods. The black line represents the correlation between the two inventory methods (Spearman's rank correlation ρ = 0.320). *Am. maculatum*: *Ambystoma maculatum*; *Am. laterale*: *Ambystoma laterale*; *An. americanus*: *Anaxyrus_americanus*; *He. scutatum*: *Hemidactylium_scutatum*; *Hy. versicolor*: *Hyla versicolor*; *L. catesbeianus*: *Lithobates catesbeianus*; *L. clamitans*: *Lithobathes clamitans*; *L. palustris*: *Lithobates palustris*; *L. pipiens*: *Lithobates pipiens*; *L. sylvaticus*: *Lithobates sylvaticus*; *N. viridescens*: *Notophthalmus viridescens*; *Pl. cinereus*: *Plethodon cinereus*; *Ps. crucifer*: *Pseudacris crucifer*.

## 4. Discussion

All inventory methods were able to detect amphibian species. Aggregated results of the traditional methods reported a similar species richness to eDNA sampling, but the species composition differed between the methods. Most detected amphibians had already been observed in the area using the traditional inventory methods. However, and as pointed out in previous studies, eDNA and the traditional method do not have the same detection success with every species present [16].

The observed discrepancy partly originates from the wide ecological and behavioral range of amphibians, but also from the technical limitations of the inventory methods. Indeed, while all amphibians reported are associated to vernal pools, the strength of their association varies. eDNA amplification was the only approach detecting *Am. laterale*. This fossorial salamander, which commonly shelters in forest liter and under rocks, is typical of deciduous forests and vernal pools from the Great Lakes region and Quebec. This species only uses vernal pools during the breeding season, where it lays eggs in the early spring, which explains its detection using eDNA. Similarly, several species, notably *Notophthalmus viridescens* (Rafinesque, 1820) and *Am. maculatum*, were detected more frequently with eDNA metabarcoding than with the traditional approaches. The former uses vernal pools and its surroundings as a foraging habitat and dispersal stepping stones (only adults are aquatic, juveniles roam the forest litter), but also sometimes as breeding grounds [29,51]. The latter, even though strictly terrestrial, relies on vernal pools to lay its eggs throughout spring, and several salamander egg clusters were observed. This active use of vernal pools, especially for reproduction, explains their common eDNA detection: large DNA quantities were released during the sampling period. The traditional sampling methods mostly failed to assess the real occurrence of *N. viridescens* and *Am. maculatum*, but this ought to be expected. Gee traps or acoustic recorders are not optimal as these species are terrestrial (for the juvenile phase of *N. viridescens*), and they do not chorus to attract mates. For Anura, tadpole trapping could be a potential solution, but taxonomic assignment is difficult due to the intra and inter species' morphological plasticity [52–54]. Without using genetic barcoding, only two main methods are available to assign a species name to a tadpole: both collecting and rearing spawn from an identified mating pair of frogs or keeping tadpoles alive until their metamorphosis; neither of these solutions are optimal for recurrent population surveys. For Urodela, aquatic larvae taxonomic assignment is easier, but it would require dedicated traps with a smaller mesh size and identification in the field. Conversely, eDNA failed to reliably detect common species in the area, notably *Pl. cinereus* and *Ps. crucifer*. While observed in the vicinity of all sampled vernal pools, *Pl. cinereus* lives and reproduces in the forest litter, and chances to detect its DNA in vernal pools are intuitively limited. The low eDNA detection rate of *Ps. crucifer* is surprising. This species also lives in the litter but uses vernal pools for reproduction from March to June [54,55]. Using acoustic receivers (which cover areas orders of magnitude larger than the sampled vernal pools), its chorus could be heard over several hundred meters (pers. obs.). A plausible explanation is that *Ps. crucifer* did not reproduce in most sampled vernal pools only by chance. While two common amphibian species previously reported in the area were not detected here (*L. septentrionalis* and *E. bislineata*), our results confirm other studies' results: the traditional methods and Eukaryote eDNA metabarcoding can perform amphibian diversity surveys around vernal pools [35]. Most importantly, our results agree with previous studies conducted in different systems [16,35] that underlined the difference in communities' composition and species detection rate depending on the inventory approach considered, as well as the higher repeatability of eDNA metabarcoding.

While eDNA and the traditional survey methods used here detect almost the same absolute number of species, the communities detected with eDNA were significantly more similar than those using the traditional methods. A sampling artifact could explain this result, but all fieldwork was conducted in May–June 2019, eliminating yearly variations of amphibian distributions or abundance. This important result has at least two explanations. First, DNA can stay up to three weeks in freshwater bodies [56], with variations based on

environmental conditions [57]: species seldomly using vernal pools can be detected over a large time window [33]. The second explanation is related to this window of detection, as active search surveys were conducted during the day over short periods, and thus, limiting the chances of detecting species (underlined by the Chao2 index of the traditional methods showing that a higher proportion of species were detected once). This result reveals that eDNA has several advantages compared to the traditional methods, even considering the limitations related to species behavior and ecology. Indeed, active search surveys were highly variable and required previous knowledge of species' identification characters. In addition, the time needed to collect the eDNA samples was trivial compared to setting the Gee traps, acoustic recorders, and conducting active search surveys. Furthermore, the time and expertise required to retrieve and analyze the acoustic data (tens of hours) needs to be accounted for. Field time and labor are thus significant cost barriers to amphibian surveys, and eDNA metabarcoding's repeatability and lower overall cost can represent a significant asset when the limits are first assessed, in agreement with the other studies [16]. Once the eDNA metabarcoding protocol was developed, the laboratory work and data analysis took only days, with a cost per sample of ~CAD\$ 33 (for consumables and a Miseq run of 192 samples). Different technologies, such as quantitative or digital PCR, could help in species' specific management plans [33], but metabarcoding can be sufficient on its own or complement the traditional surveys, according to the aims of stakeholders. Regarding the system used here, eDNA could provide the general habitat use data as it reliably detects amphibian species sensitive to habitat modifications and degradations, but difficult to observe with the traditional methods (e.g., *N. viridescens*, *Am. maculatum*, and *Am. laterale* [58–60]). This could be used to test the effect of different tree-cut types and canopy openings on vernal pools.

In short, and as expected, our study underlines that a precise knowledge of species' ecology and behavior, with sampling designed accordingly, is necessary to conduct and interpret amphibian biodiversity surveys, as well as biodiversity surveys in general. Indeed, the eDNA methods do not provide information on population size, condition, developmental stage, sex, nor allow for the tagging or sampling of the target animals, which makes it difficult to interpret changes in detection over time without a priori information [39]. However, the traditional methods also have limitations: acoustic recorders give a biased representation of a population status, as only males chorus, while access to the sex ratio data is important as amphibians' sex differentiation is partly based on the environmental factors (e.g., temperature and chemical compounds [61–63]). Therefore, we have shown, in line with the previous studies [16], that not a single tool can detect all species at once and eDNA metabarcoding complements the traditional survey methods, especially for litter-dwelling species.

## 5. Conclusions

Temperate forests are facing major biodiversity losses, mainly due to human activities, requiring intensive biodiversity inventories to monitor these losses. This study represents another important step in the development and application of eDNA metabarcoding for the detection and monitoring of forest amphibians. When comparing the results of eDNA metabarcoding with the traditional inventory methods, we found that species' ecology and behavior is a key factor of its detectability, even if they are present in the vicinity of the sampled water body. While eDNA has inherent limitations, as animals are not seen and counted, this approach is comparatively inexpensive and has a higher repeatability, while also being non-invasive. It seems to be particularly useful to detect cryptic species such as Urodela, while Anura are generally best detected using other methods. We conclude that eDNA sampling should be considered for integration as a standard monitoring tool, after a first assessment of amphibians' diversity using the traditional methods. eDNA metabarcoding, due to its low field-cost and fast data analysis, has a potential for replication and geographic scalability that the traditional methods cannot match without unlimited financial resources.

**Supplementary Materials:** The following supporting information can be downloaded at: https://www.mdpi.com/article/10.3390/f14101930/s1, Text S1: Usearch parameters; Figure S1: zOTUs' accumulation curves of the merged CO1 metabarcodes; Table S1: Summary of sampling effort; Table S2: Presence–absence of each amphibian species per sampled vernal pool and inventory method.

**Author Contributions:** Conceptualization, A.D. and Y.S.-G.; methodology, A.D., E.L. and Y.S.-G.; formal analysis, B.P. and E.L.; data curation, E.L. and Y.S.-G.; writing—original draft preparation, B.P.; writing—review and editing, B.P., A.D., E.L. and Y.S.-G.; supervision, A.D. and Y.S.-G.; project administration, A.D. and Y.S.-G.; funding acquisition, A.D. and Y.S.-G. All authors have read and agreed to the published version of the manuscript.

**Funding:** This research was funded by the Natural Sciences and Engineering Research Council of Canada (CRD grant # CRDPJ 506241-2016) with additional financial support from Kenauk Canada, Inc., Nature Conservancy of Canada, and the Sustainable Forestry Initiative.

**Data Availability Statement:** The raw sequencing reads will be submitted to the NCBI SRA database upon acceptance. The scripts for the bioinformatic data analysis are available as supplementary information.

**Acknowledgments:** Protocols involving the trapping and handling of amphibians were approved by the Comité institutionnel de protection des animaux de l'Université du Quebec à Montréal (protocol # 0318-938-0319), and a scientific permit to collect amphibians was provided by the Ministère des Forêts, de la Faune et des Parcs du Québec (#PM_19-07-SF-004-GR-0). We thank Florence Tauc, Frédéric Moore, Clotilde Pires, and Antoine Berthoux for their help collecting the data in the field, Laurence Danvoye for her help in processing the samples in the lab, and the personnel of the Centre d'expertise et de services Génome Québec for the sequencing of the samples. We thank Matthew McCormack for proofreading this manuscript. We thank Liane Nowell from Kenauk Institute for granting access to the Kenauk nature reserve.

**Conflicts of Interest:** The authors declare no conflict of interest. The funders had no role in the design of the study; in the collection, analyses, or interpretation of the data; in the writing of the manuscript; or in the decision to publish the results.

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
