# Peer review of "Vernal Pool Amphibian Inventories in the Temperate Forests of Eastern North America: Can Environmental DNA Replace the Traditional Methods?"

_forests, doi:10.3390/f14101930_

Round 1

Reviewer 1 Report

My comments and suggestions are in the attached file.

Author Response

Page 2, line 89: “for this area, is altitude important?”

The Kenauk Reserve is situated in southern Quebec, close to the Saint Laurent River, and altitudinal gradient are not pronounced in the area. Most topographic features do not exceed 400m, and forest cover is continuous (except for a few clear cuts). Altitude is not expected to influence our results, but we included a gross estimate of its range in the text, based on freely available topographic data.

Page 3, line 91: “In figure 1, what are these codes? I think they should be mentioned in the legend.”

The codes represent the names associated with each vernal pool sampled for this study. We thank the reviewer for pointing out that this information was not included in the legend as it should have. We modified the legend accordingly.

Figure 1: Positions of sampled vernal pools in Kenauk Reserve (QC, Canada). Vernal pools’ names are also indicated. Based on [29]

Page 3, lines 99 and 106: Supplementary material Figure S1? Text S1? Table S1?

We thank the reviewer for pointing out this mistake. We were referring to Table S1. The text was corrected accordingly.

Page 4, line 135: “Are these primers universal?” (regarding MHemF and dgHCO2198)

The primers combination MHemF and dgHCO2198 amplifies a wide range of metazoan (arthropods, vertebrates, annelids, and mollusks), but they preferentially target arthropods. The reverse primer (dgHCO2198) is taken from Meyer et al. (2005), which is an improvement from the universal metazoan PCR primers designed by Folmer et al. (1994). The forward primer (MHemF; Park et al. 2011) amplifies a smaller fragment of the CO1 and is especially useful in arthropods. This is the reason why we used a second set of primers. These details are available in the reference cited in the manuscript [34].

Meyer CP, Geller JB, Paulay G. 2005. Fine scale endemism on coral reefs: archipelagic differentiation in turbinid gastropods. Evolution. 59(1):113–125.

Folmer O, Black M, Hoeh W, Lutz R, Vrijenhoek R. 1994. DNA primers for amplification of mitochondrial cytochrome c oxidase subunit I from diverse metazoan invertebrates. Mol Mar Biol Biotechnol. 3(5):294–299.

Park, D. S., Foottit, R., Maw, E., & Hebert, P. D. (2011). Barcoding bugs: DNA-based identification of the true bugs (Insecta: Hemiptera: Heteroptera). Plos one, 6(4), e18749.

Page 5, line 167: “is there a reference for this software (regarding Illumina Miseq)?”

We could not find a reference for the Illumina Miseq software, it is preinstalled on the sequencer and we receive the samples already demultiplexed from the sequencing service provider. We found however a reference describing the complete Miseq workflow. We included the reference to the manuscript.

Page 5, line 179: “In Table S2, what means are 1 and 0 numbers? I think they should be written to table legend”

We thank the reviewer for pointing out this mistake. The numbers 1 and 0 mean presence or absence, respectively. The legend was corrected accordingly.

Page 5, line 204: Was it mentioned from NGS in the material and method part? If yes, what is this analysis the open state?

NGS refers to Next Generation Sequencing, the generic term referring to the metabarcoding approach used in this study. We changed the title of this section to prevent any confusion, as this abbreviation was not introduced earlier in the text.

Reviewer 2 Report

To amphibians that are occupying seasonal habitats, a reliable approach of inventory for these species is critical. Therefore, I supported the acceptance of the manuscript, largely for its value in improving the terrestrial amphibians monitoring.  For the abstract, please highlight clearly the exceptional supports that eDNA can offer over the traditional survey method for instance in in this case, consistent detection on vernal pools' inhabitants or breeders,  increase the resolution of detection on non-singing, terrestrial amphibians and also the adding accuracy on detection of amphibians with low phenotypic variation. A very brief line of explanation about how behaviour and ecology play the integral role in influencing current eDNA inventory is also necessary in the abstract. 

As for the writing, the clarity is decent, except that rephrasing would be needed for some lengthy sentences. 

Author Response

To amphibians that are occupying seasonal habitats, a reliable approach of inventory for these species is critical. Therefore, I supported the acceptance of the manuscript, largely for its value in improving the terrestrial amphibians monitoring.  For the abstract, please highlight clearly the exceptional supports that eDNA can offer over the traditional survey method for instance in in this case, consistent detection on vernal pools' inhabitants or breeders,  increase the resolution of detection on non-singing, terrestrial amphibians and also the adding accuracy on detection of amphibians with low phenotypic variation. A very brief line of explanation about how behaviour and ecology play the integral role in influencing current eDNA inventory is also necessary in the abstract.

We thank the reviewer for their comments. As requested, we tried to underline even more the potential of eDNA metabarcoding to monitor vernal pools’ amphibians. However, the length of the abstract is limited, and we must make a selection on the highlighted results.

As for the writing, the clarity is decent, except that rephrasing would be needed for some lengthy sentences.

Based on this comment, we asked an English native speaker to proofread the manuscript (Matthew McCormack, thanked in the Acknowledgments). He did not find any particularly problematic sentence, so we decided to keep the text as it is.

Reviewer 3 Report

I read the paper titled “Vernal pool amphibian inventories in temperate forests of east-2 ern North America: can environmental DNA replace traditional 3 methods?”. It is well written and interesting affording an important topic on the assessment of the use of temporary ponds by amphibians using traditional monitoring and eDNA samplings.

The study compared 12 pools. Methods seem consistent with results and the paper deserves publication.

Only some minor points need to be solved prior to publication:

In Methods section, the information that surveys occurred between May_June of the year 2019 comes after that specific dates are reported. It could be better to have from the beginning the year and the indication of the whole period in which the study took place.

I am also concerned about spatial autocorrelation in the data collected between field sites. Please check that spatial autocorrelation did not exist in the data set maybe using a Moran's I tests on residuals of models

Lines 45-46 see as an example Ficetola et al.,  2019. Environmental DNA and metabarcoding for the study of amphibians and reptiles: species distribution, the microbiome, and much more. Amphibia-Reptilia, 40(2), 129-148.

Tables’ captions should be auto-explicative. Better to write the explanations of the abbreviations used.

Caption of Fig.3; I guest it should be “Amphibians”.

Lines 381-382, the end of the sentence of conclusions is a bit cryptic. Please try to be more explicative.

Author Response

I read the paper titled “Vernal pool amphibian inventories in temperate forests of eastern North America: can environmental DNA replace traditional methods?”. It is well written and interesting affording an important topic on the assessment of the use of temporary ponds by amphibians using traditional monitoring and eDNA samplings.

The study compared 12 pools. Methods seem consistent with results and the paper deserves publication.

Only some minor points need to be solved prior to publication:

In Methods section, the information that surveys occurred between May_June of the year 2019 comes after that specific dates are reported. It could be better to have from the beginning the year and the indication of the whole period in which the study took place.

We do not understand this comment and we would like the reviewer to clarify this point. In the Methods, we wrote that the study took place between May and June 2019 (end of the 2.1. Sampling site subsection) and then indicate specific dates down the text for each sampling method (in the 2.2. Traditional inventory of amphibian diversity and 2.3. eDNA collection and processing subsections). We did not modify the text.

I am also concerned about spatial autocorrelation in the data collected between field sites. Please check that spatial autocorrelation did not exist in the data set maybe using a Moran's I tests on residuals of models.

We agree that spatial autocorrelation could be a problem if the aim of the study was to assess species diversity at a pond scale, comparing vernal pools. However, this is not the point of this study as we are interested in comparing methods, not the sites and we do not use spatial data.

However, to test for potential correlation between beta-diversity and geographic distances, we performed two Mantel tests (one for eDNA inventories and one for traditional sampling) comparing the Sorensen similarity and geographic distances between pairs of vernal pools, performing 9999 permutations to assess significance. Neither eDNA methods (Mantel statistic r: -0.02359, pvalue= 0.5703) nor traditional sampling (Mantel statistic r: 0.1321, pvalue = 0.0928) presented a correlation between sites geographic distance and their species composition. This was not included in the manuscript as it was not the goal of this study.

Lines 45-46 see as an example Ficetola et al.,  2019. Environmental DNA and metabarcoding for the study of amphibians and reptiles: species distribution, the microbiome, and much more. Amphibia-Reptilia, 40(2), 129-148.

We thank the reviewer for this suggestion and we included the reference in the text.

Tables’ captions should be auto-explicative. Better to write the explanations of the abbreviations used.

We could not find which abbreviations the reviewer is referring to. All abbreviations used in Table 1 are explained. The abbreviation of latin species names is used only after the full name has already been introduced. The potential missing abbreviations are the arbitrary sites names in Table S2, as pointed out by the first reviewer in Figure 1. We specified that site names are arbitrary in the captions of both Figure 1 and Table S2.

Caption of Fig.3; I guest it should be “Amphibians”.

We thank the reviewer for pointing out this mistake. We corrected the caption.

Lines 381-382, the end of the sentence of conclusions is a bit cryptic. Please try to be more explicative.

We developed the last sentence to clarify the point we want to convey:

eDNA metabarcoding, due to its low field-cost and fast data analysis, has a potential for replication and geographic scalability that traditional methods cannot match without unlimited financial resources.